# COVID-19 Infection in Pregnancy: PCR Cycle Thresholds, Placental Pathology, and Perinatal Outcomes

**DOI:** 10.3390/v13091884

**Published:** 2021-09-21

**Authors:** Estibalitz Laresgoiti-Servitje, Jorge Arturo Cardona-Pérez, Rosa Gabriela Hernández-Cruz, Addy Cecilia Helguera-Repetto, María Yolotzin Valdespino-Vázquez, Elsa Romelia Moreno-Verduzco, Isabel Villegas-Mota, Sandra Acevedo-Gallegos, Mario Rodríguez-Bosch, Moisés León-Juárez, Mónica Aguinaga-Ríos, Irma Coronado-Zarco, Alejandro Ortiz-Calvillo, María Antonieta Rivera-Rueda, Carolina Valencia-Contreras, María de Lourdes Gómez-Sousa, Mario Solis-Paredes, Juan Carlos Rodriguez-Aldama, Rafael Galván-Contreras, Ricardo Figueroa-Damián, Manuel Cortés-Bonilla, Guadalupe Estrada-Gutierrez, Salvador Espino-y-Sosa, Claudine Irles

**Affiliations:** 1Clinical Sciences, School of Medicine and Health Sciences, Tecnológico de Monterrey, Mexico City 64849, Mexico; estibalitz.laresgoiti@tec.mx; 2Dirección General, Instituto Nacional de Perinatología, Mexico City 11000, Mexico; cardona@inper.gob.mx; 3Departamento de Fomento y Herramientas Educativas, Instituto Nacional de Perinatología, Mexico City 11000, Mexico; cruzroga33@gmail.com; 4Departamento de Immunobioquímica, Instituto Nacional de Perinatología, Mexico City 11000, Mexico; addy.helguera@inper.gob.mx (A.C.H.-R.); moisesleoninper@gmail.com (M.L.-J.); 5Departamento de Anatomía Patológica, Instituto Nacional de Perinatología, Mexico City 11000, Mexico; yolotzv@gmail.com; 6Subdirección de Servicios Auxiliares de Diagnóstico, Instituto Nacional de Perinatología, Mexico City 11000, Mexico; elsamover@yahoo.com; 7Departamento de Enfermedades Infecciosas e Epidemiología, Instituto Nacional de Perinatología, Mexico City 11000, Mexico; isavillegas13@outlook.com (I.V.-M.); j_carlos128@gmail.com (J.C.R.-A.); rafagcsx@gmail.com (R.G.-C.); 8Departamento de Medicina Materno-Fetal, Instituto Nacional de Perinatología, Mexico City 11000, Mexico; dracevedo_sandra@yahoo.com.mx; 9Subdirección de Ginecología e Obstetricia, Instituto Nacional de Perinatología, Mexico City 11000, Mexico; mario.rodriguez@inper.gob.mx; 10Coordinación de Genética Clínica, Departamento de Medicina Materno-Fetal, Instituto Nacional de Perinatología, Mexico City 11000, Mexico; aguinagamonica09@gmail.com; 11Subdirección de Neonatología, Instituto Nacional de Perinatología, Mexico City 11000, Mexico; irma.coronado@inper.gob.mx; 12Subdirección de Investigación Clínica, Instituto Nacional de Perinatología, Mexico City 11000, Mexico; aortizcalvillo@gmail.com (A.O.-C.); salvadorespino@gmail.com (S.E.-y.-S.); 13Unidad de Cuidados Intensivos del Recién Nacido, Instituto Nacional de Perinatología, Mexico City 11000, Mexico; marivera1309@yahoo.com.mx; 14Unidad de Cuidados Intermedios del Recién Nacido, Instituto Nacional de Perinatología, Mexico City 11000, Mexico; dra.cvalencia@gmail.com; 15Departamento de Posgrado e Investigación, Instituto Nacional de Perinatología, Mexico City 11000, Mexico; lulugomezsousa@gmail.com; 16Departamento de Genética y Genómica Humana, Instituto Nacional de Perinatología, Mexico City 11000, Mexico; juan.mario.sp@gmail.com; 17Departamento de Infectología e Inmunología, Instituto Nacional de Perinatología, Mexico City 11000, Mexico; rfd6102@yahoo.com.mx; 18Dirección Médica, Instituto Nacional de Perinatología, Mexico City 11000, Mexico; manuel.cortes@inper.gob.mx; 19Dirección de Investigación, Instituto Nacional de Perinatología, Mexico City 11000, Mexico; gpestrad@gmail.com; 20Departamento de Fisiología y Desarrollo Celular, Instituto Nacional de Perinatología, Mexico City 11000, Mexico

**Keywords:** SARS-CoV-2, pregnancy, hypertensive disorder, placenta

## Abstract

(1) This study aimed to evaluate characteristics, perinatal outcomes, and placental pathology of pregnant women with or without SARS-CoV-2 infection in the context of maternal PCR cycle threshold (C_T_) values. (2) This was a retrospective case-control study in a third-level health center in Mexico City with universal screening by RT-qPCR. The association of COVID-19 manifestations, preeclampsia, and preterm birth with maternal variables and C_T_ values were assessed by logistic regression models and decision trees. (3) Accordingly, 828 and 298 women had a negative and positive test, respectively. Of those positive, only 2.6% of them presented mild to moderate symptoms. Clinical characteristics between both groups of women were similar. No associations between C_T_ values were found for maternal features, such as pre-gestational BMI, age, and symptomatology. A significantly higher percentage of placental fibrinoid was seen with women with low C_T_s (<25; *p* < 0.01). Regarding perinatal outcomes, preeclampsia was found to be significantly associated with symptomatology but not with risk factors or C_T_ values (*p* < 0.01, aOR = 14.72). Moreover, 88.9% of women diagnosed with COVID-19 at <35 gestational weeks and symptomatic developed preeclampsia. (4) The data support strong guidance for pregnancies with SARS-CoV-2 infection, in particular preeclampsia and placental pathology, which need further investigation.

## 1. Introduction

In December 2019, an outbreak of a coronavirus that causes severe acute respiratory syndrome (SARS-CoV-2) was observed in Wuhan, China. The virus causes the widely known Coronavirus disease (COVID-19) [1], with more than 123 million cases in 223 countries and more than 2 million deaths worldwide. Pregnant women are a high-risk population, especially vulnerable to infectious diseases due to the immune system’s delicate regulation during the gestational period [2] and the risk of vertical transmission of the infectious disease to the fetus [3]. The previously mentioned changes in the immune system during pregnancy may lead to higher mortality rates related to viral infections than the general population [4].

During the COVID-19 pandemic, several research groups worldwide have reported the clinical characteristics of pregnant women infected with SARS-CoV-2 [5], the virus’s ability to infect the placenta and the presence of vertical transmission from mother to child [6,7,8]. The histopathological spectrum of placentas from maternal–neonatal dyads with SARS-CoV-2 infection showed histiocytic intervillositis and trophoblast necrosis [9,10]. The impact of COVID-19 in pregnancy is yet to be fully understood. The most common symptom at the onset, which usually occurs between days 1 and 7 post-infection, is fever, followed by cough [11,12]. This large referral center for perinatal health care has one of the highest prevalences of SARS-CoV-2 among pregnant women, around 30%. However, less than 3% of patients were symptomatic [13,14]. The mild to moderate symptoms associated with SARS-CoV-2 infection were headache, dyspnea, myalgia, and olfactory/gustatory dysfunction [13].

There is still scarce information published regarding the behavior of COVID-19 in pregnant women. Most of the information available concerns the adult non-pregnant population. This study contributes to the knowledge of the behavior of COVID-19 in pregnant women in a Hispanic population.

The study aimed to assess differences in clinical aspects between pregnant women with SARS-CoV-2 positive PCR test results and those with negative results and evaluate if the cycle threshold (C_T_) values at delivery can impact the disease’s clinical presentation in relation to placental pathology and pregnancy outcomes. Lastly, this study aimed to assess if there is an association of preterm birth or preeclampsia in pregnant women with COVID-19 compared to those without the disease.

## 2. Materials and Methods

This study aimed to report the National Institute of Perinatology (INPer, Instituto Nacional de Perinatología Isidro Espinosa de los Reyes) experience in Mexico City during the COVID-19 pandemic and to evaluate the behavior of SARS-CoV-2 infection in pregnant women who attended a large, third-level referral institution. This observational case-control study was accepted by the Institutional Research and Ethics Committees of the INPer, grants number 2020-1-32 and 2020-1-31, and followed STROBE guidelines.

The analyzed database comprises pregnant women who were routinely screened for COVID-19 during their pregnancy appointments for delivery or an obstetric emergency from mid-April 2020 to mid-September 2020. The data included a RT-qPCR test as well as clinical and laboratory results.

This study assessed differences in clinical aspects, symptomatology, and outcomes between pregnant women with positive SARS-CoV-2 RT-qPCR test results and those with negative results. It also evaluated if the qualitative PCR C_T_ values were associated with placental pathology and adverse outcomes at delivery.

### 2.1. Research Design and Approach

This study is a retrospective cohort of 1126 pregnant women who attended this large referral health care center from mid-April 2020 to mid-September 2020. The inclusion criteria were: women in the third trimester of pregnancy with a negative or positive PCR test result for SARS-CoV-2 by RT-qPCR; age range between 12 and 45 years old; women from all socioeconomic status and racial origin were included in this study. Universal SARS-CoV-2 testing of nasopharyngeal/oropharyngeal swabs was performed on all pregnant women attended at the INPer, upon arrival at TRIAGE. The time interval between the RT-qPCR test and delivery was 24–48 h depending on whether the pregnant woman was admitted for an appointment for delivery, i.e., a programmed visit 24–48 h before delivery, or if it was an obstetric emergency, in which case the test was performed upon admittance and the test result obtained between the next 24 h. The pathological examination of the placenta was performed by an experienced perinatal pathologist (Y.V-V) on the day of delivery, at the same interval of time as the RT-qPCR test. The report includes the clinical history of the patients, as well as macroscopic (weight, measurements, and appearance of the parenchyma) and microscopic (description of tissue and pathological findings, including the presence of placental dysmaturity, vasculopathy, fibrinoid, chorangiosis, chorioamnionitis, hemorrhage, or infarction) description. However, not all placentas were examined. Submitting all placentas for pathological examination is not clinically indicated. The placental examination was recommended for women with a positive RT-qPCR test. However, not all of the placentas were routinely obtained and sent to the Department of Anatomic Pathology. Furthermore, some of the placentas evaluated by the pathologist also had a RT-qPCR test result.

The Clinical Research Direction team collected the data at the INPer. To maintain the participants’ confidentiality, all women enrolled in the study were assigned a study number, and the names of the participants were not disclosed. This study did not pose any potential physical or psychological harm to the participants. Each woman signed an informed consent form that guaranteed confidentiality to the participant.

### 2.2. Reverse Transcription Quantitative Real-Time Polymerase Chain Reaction (RT-qPCR) for SARS-CoV-2

Nasopharyngeal/oropharyngeal swabs were performed following a standardized procedure. RNA was purified using the RNeasy Qiagen commercial kit (Qiagen, Hilden, Germany). The SARS-CoV-2 RNA was detected following La Charité, Berlin protocol, using TaqPath 1 step RT-PCR master Mix, CG commercial kit (Thermo Fisher Scientific, Waltham, MA, USA), and probes and primers designed for RdRP and E viral genes. RNase P human gene was used as RNA isolation control. RT-qPCR was performed on a StepOnePlus instrument (Applied Biosystems/Thermo Fisher Scientific). Each 20 µL RT-PCR reaction contained 5 µL of RNA, 5 µL enzyme mix, primers, and probes. Conditions of the thermocycler followed the La Charité protocol. A cycle threshold (C_T_) value <37.5 was considered positive for SARS-CoV-2 RNA. C_T_ values were collected using the threshold at 0.035 fluorescence level.

### 2.3. C_T_ Values

In this study, the term viral load will not be considered, as the viral assessment was not performed using a quantitative PCR test that permits to evaluate the number of RNA copies per ml. Instead, the PCR cycle threshold (C_T_) values will be reported. This means that a standard curve was not constructed for these PCR tests. In order to report RNA viral loads quantitatively, a standard plot should be built with several serial dilutions of copies of cRNA plotted on the *x*-axis and the C_T_ values plotted on the *y*-axis [15,16]. Thus, the PCR results in this study cannot be considered a quantitative measure of viral load but a more qualitative appreciation. Lower C_T_ values may be related to a higher viral load, although there is no perfect correlation [17,18], and this is the reason why C_T_ should not be considered equivalent to viral load. In this study, a C_T_ value <37.5 was deemed to be positive, and C_T_ counts were categorized as high (C_T_ > 30), medium (C_T_ of 25–30), or low (C_T_ < 25) according to cut-off points used in previous studies [19].

### 2.4. Statistical Analysis

The dataset was summarized, checked for outliers, and a missing value analysis was executed. No pattern was identified in missing values. Descriptive statistics were evaluated, and means and standard deviations are reported for numerical values. Numbers and percentages are reported for categorical variables. To assess differences between the group’s clinical and maternal characteristics, a chi-square or Fisher test was reported depending on the number of cases per cell. Differences between groups in quantitative variables were evaluated with Student *t*-tests. A stepwise logistic regression model was fit to evaluate which COVID-19 manifestations were most likely to be associated with a positive PCR test result. A chi-square automatic interaction detector (CHAID) tree was constructed as a sensitivity analysis for the logistic regression. Analysis of variance was executed to assess the difference in the values of PCR C_T_ between the reason women were tested for COVID19 and between pre-pregnancy body mass index categories. Analyses of covariance were also executed, controlling for maternal age and pre-pregnancy BMI. Lastly, forward logistic regression analysis and decision trees were fitted to assess the association of preeclampsia and preterm birth with maternal variables in women with and without COVID-19. Bootstrapping with 1000 samples was executed for both regression models. QUEST decision trees were performed as sensitivity analyses.

Analyses were performed using SAS University Edition (Cary, NC, USA), RStudio environment for R, Version 1.1.419, and SPSS version 27 (IBM Corp., Armonk, NY, USA).

## 3. Results

### 3.1. Clinical Characteristics, C_T_s, and Placental Pathology

#### 3.1.1. Maternal Demographic and Clinical Characteristics

In total, 1126 pregnant women were included in this retrospective cohort study, of whom 828 had a negative test, and 298 had a positive test. All patients were attended at the National Institute of Perinatology and were tested for COVID-19 because they had a programmed visit for delivery (42.3%), an obstetric emergency (49.9%), or were hospitalized for pregnancy resolution and suspected to have COVID-19 (6.9%). Seventy-three percent of women tested had a negative result (CI, 70.6–76.1%). Only 2.6% of tested patients had COVID-19 mild symptomatology (CI, 1.8–3.6%), whereas 23.9% of women had a positive test and were asymptomatic (CI, 21.4–26.4%).

Mean maternal age was 28 years old (SD 7.27), and mean gestational week at delivery was 33.2 (SD 8.3) weeks. None of the women included in this study died from COVID-19. Most patients (61%) had a pre-pregnancy BMI that characterized them as overweight (BMI 25–29.9) or obese (BMI ≥ 30). The most common chronic disorders present in these patients were diabetes, followed by hypertension and autoimmunity. Many patients with asthma (68.8%) and hypothyroidism (51.4%) were positive for COVID-19 during pregnancy. Positive tests were significantly higher in women with hypothyroidism. Maternal characteristics are shown in Table 1.

There were no differences in temperature measurements between patients with positive and negative results. However, breathing rate, heart rate, systolic blood pressure (SBP), and diastolic blood pressure (DBP) were significantly higher in COVID-19 positive patients (*p* < 0.05). The majority of patients were tested when evaluated for obstetric emergency (562), the rest were evaluated during their programmed control visits 24–48 h prior to delivery and were asymptomatic (456), and only 20 patients were symptomatic during their programmed visits for delivery, of whom eight were positive for COVID-19. After being tested, most of the patients were admitted to the hospital (76.4%), primarily related to their obstetric condition and not related to COVID-19 symptomatology. Table 2 shows the clinical characteristics of patients at triage. Hence, 90.3% of pregnant women with a positive PCR test were asymptomatic, and only 9.7% were symptomatic.

#### 3.1.2. PCR Test C_T_s

This study’s theoretical basis comes from the knowledge that lower C_T_ values have been related to severe disease and that increased BMI and age can be associated with adverse COVID19 outcomes [1,16,18]. Figure 1 shows no differences in age between patients with positive and negative results to COVID19 during pregnancy (a). Overweight women tended to have lower C_T_ values, and women with class III obesity had a lower gestational age at triage (b).

A stepwise logistic regression model and a decision tree were fitted to evaluate which COVID-19 manifestations commonly reported in non-pregnant populations (cough, fever, headache, dyspnea, myalgias, rhinorrhea, diarrhea, thrombosis, neurological symptoms, arrhythmia, lethargy, Appendix A) were associated with PCR positivity in pregnant women.

The model was significant (*p* = 0.0027) with an AIC = 1284. The only clinical variable significantly associated with PCR positivity in pregnant women was cough (*p* = 0.008, OR = 17.28, CI 2.072–144). The model correctly predicted 74.4% of cases. None of the other variables were significantly associated with a positive PCR test. The estimates for the regression are shown in Appendix A in Appendix A. Only the variable cough was able to split the decision tree, with 87.5% of patients with cough having a positive test, whereas only 26% of patients without cough had a positive test. Figure 2 shows tree cross-validation (a) and regression tree (b), and CHAID tree (c).

Analysis of variance was executed to evaluate the difference in SARS-CoV-2 PCR C_T_ values between symptomatic and asymptomatic pregnant patients and between pre-pregnancy BMI categories.

There were no significant differences in C_T_ values between symptomatic patients (33.9 ± 4.1) and asymptomatic patients (34.1 ± 5.8) (F (1291) = 0.702, *p* = 0.403), neither between symptomatic patients who had programmed visits (30.9 ± 6.2), nor asymptomatic patients (32.8 ± 4.3), those with an obstetric emergency (32.8 ± 4.1), and those who were hospitalized and suspicious to have COVID19 (32.9 ± 5.5). F (3287) = 0.51, *p* = 0.67. Chi-square analyses were used to evaluate if the presence of three C_T_ values: low (C_T_ > 30), medium (C_T_ 25–30), or high (C_T_ < 25), could be associated with symptomatology in COVID-19 patients. There were no significant differences between symptomatic and asymptomatic women’s C_T_ values (*p* = 0.214). High C_T_ values were found in 84.5% (*n* = 223) of asymptomatic women compared to 82.1% (*n* = 23) of symptomatic COVID-19 patients, while a trend was found for low C_T_s in 14.3% (*n* = 4) of symptomatic women compared to 6.4% (*n* = 17) of asymptomatic COVID-19 patients. No significant differences were found between C_T_ categories and the presence or absence of symptomatology in a chi-square analysis.

A two-way ANOVA that evaluated differences in C_T_ values between symptomatic and asymptomatic patients and the interaction with pre-pregnancy BMI categories did not show any differences between symptomatic or non-symptomatic patients (*p* = 0.80), between pre-pregnancy BMI categories (*p* = 0.62), nor between the interaction (*p* = 0.84), as shown in Figure 3.

No statistical differences were found in C_T_ counts between symptomatic and asymptomatic pregnant women after controlling for maternal age and pre-pregnancy BMI in the analysis of covariance (F = 0.463. *p* = 0.63) (Appendix A).

#### 3.1.3. Placental Pathology and C_T_s

From the 298 COVID-19 positive pregnant women in this study, sixty-six placentas were evaluated by an expert pathologist for the presence of placental dysmaturity, vasculopathy, fibrinoid, chorangiosis, chorioamnionitis, hemorrhage, or infarction. Sixty-two (93.9%) placentas were from live births, two (3%) were from miscarriages, and two (3%) were from stillbirths. Forty-nine (74.2%) of the placentas were from mothers with C_T_s > 30, 6 (9.1%) were from mothers with medium C_T_ values, and 11 (16.7%) were from women with low C_T_s < 25. Placental weight percentiles were calculated according to gestational age at delivery. The mean percentile was 14.73 SD 19.44, with a median of three; placentas were in their majority small. A chi-square test revealed no significant differences between placental percentile categories and C_T_ value categories (*p* = 0.42). The time between positive PCR test and delivery of the placentas evaluated had a mean of two days. There was no significant correlation between the C_T_s and the days between diagnosis and delivery.

Thirty-two out of the 66 placentas were evaluated for SARS-CoV-2 by RT-qPCR. Fifteen tested positive with C_T_ values higher than 29.5. A ROC curve was executed with the maternal PCR C_T_ values to obtain a cut-off point for placental positivity. The AUC was 0.618 (CI 95% 0.40–0.82). The cut-off point was 33 C_T_ (with a sensitivity of 67% and a specificity of 88%). A Fisher’s exact test showed a significant association (*p* = 0.021) between maternal C_T_ values <33 with a positive placental test (80% *n* = 8 vs. 20% *n* = 2), and a negative association when C_T_ values were ≥33 (68% *n* = 15 vs. 32% *n* = 7).

Chi-square analyses were used to evaluate if the presence of a low (<25), medium (25–30), or a high (>30) PCR C_T_ value could be associated with placental pathology.

Only the percentage of placental fibrinoid was significantly higher (*p* = 0.008) in placentas of women with C_T_ values <25, being present in 90.91% (*n* = 10) of them. The distribution of fibrinoid between groups is shown in Figure 4.

A representative image of fibrinoid is depicted in Figure 5. In the placental parenchyma, a massive deposit of perivillous fibrin deposition was observed with replacement of the exchange zone by this deposit.

Differences in percentages of C_T_ values in women who developed placental fibrinoid were also assessed according to the presence of COVID-19 symptomatology or not. Women who had a positive PCR with C_T_ counts <25 but were asymptomatic had a significantly higher presence of fibrinoid (90% (*n* = 9), *p* = 0.003)). In contrast, there were no significant differences in fibrinoid between placentas from symptomatic women with different C_T_ values. However, it is of note that all symptomatic women with C_T_ count <30 developed placental fibrinoid.

To further assess the association of placental C_T_ values with the presence of fibrinoid of women with COVID19, we performed a ROC curve. The area under the curve (AUC) was 0.635 (CI 95% 0.50–0.76) although non-significant (*p* = 0.59). Although the number of positive placentas that had fibrinoid showed a trend with a higher number (8 vs. 5), a Fisher’s exact test did not find a significant association between placental positivity and the presence of fibrinoid (*p* = 0.049).

### 3.2. Pregnancy Outcomes

Most patients delivered at term (37 ± 3.4 gestational weeks), and there were no differences between gestational age and hospitalization time at delivery between women with and without COVID-19 positivity during pregnancy. One hundred and sixty-eight women delivered preterm. The most common pregnancy resolution method was a cesarean section, followed by eutocic delivery. The most common form of preeclampsia presented by women in this study was preeclampsia with severity features. Fetal death was reported on 57 pregnancies, and COVID-19 patients had 6.4% mortality, whereas in non-COVID-19 women, fetal mortality was 4.6%. There were no differences in the birth weight or Apgar scores of children born to women who had COVID-19 during pregnancy. Table 3 shows the characteristics of patients at delivery.

Lastly, forward logistic regression analyses and decision trees were fitted to assess the association of preeclampsia and preterm birth with maternal variables in women with and without COVID-19. The logistic regression model that evaluated the association of pre-pregnancy BMI, maternal age, having COVID-19 symptomatology vs. being asymptomatic vs. COVID-19 negative test, gestational age at triage, and C_T_ count categories (low < 25, medium 25–30 and high > 30) was statistically significant (*p* = 0.008, Nagelkerke R2 = 0.37), correctly predicted 89% of cases, and showed that having at least two symptoms was significantly associated to preeclampsia compared to non-symptomatic COVID-19 pregnant women (*p* = 0.004, adjusted OR = 14.72 CI 95% 2.39–90.37). None of the other variables were significantly related to preeclampsia in this model. Logistic regression estimates and adjusted odds ratios are shown in Appendix A.

A quick, unbiased, and efficient statistical tree (QUEST) was also performed as a sensitivity analysis. The same variables in the logistic regression were considered in the decision tree. Growth was limited to minimum parent node size = 20 and minimum child node size = 5. The tree was pruned to avoid overfitting the model. The tree resulted in two splits according to the symptomatic/asymptomatic/negative variable and to gestational age at triage. The node with the highest number of preeclampsia cases was the one of symptomatic COVID-19 women, with 57.1% of cases with preeclampsia. In contrast, the node with asymptomatic women and negative PCR test only had 8.7% of cases. Most women diagnosed with COVID-19 at less than 35.43 weeks and were symptomatic developed preeclampsia (88.9%), whereas in those diagnosed after 35.43 weeks, only 33.3% developed preeclampsia. The risk estimate for this tree was 0.91 SE = 0.010. The tree is shown in Figure 6.

A binary logistic regression was fitted to estimate the association of preterm birth with pre-pregnancy BMI, maternal age, being COVID-19 symptomatic vs. asymptomatic, gestational age at triage, and C_T_ count categories (low < 25, medium 25–30 and high > 30) and the development of preeclampsia. The model was statistically significant (*p* < 0.001, Nagelkerke R2 = 0.76) and correctly predicted 94.4% of cases. Only two variables were significantly related to preterm birth: gestational age at triage and preeclampsia. Preeclampsia was positively associated with preterm birth (*p* = 0.001, adjusted OR = 24.01 CI 95% 1.44–399), whereas gestational age at triage (COVID-19 diagnosis) was negatively related to preterm birth (*p* = 0.001 adjusted OR = 0.51 CI 95% 0.34–0.76). Logistic regression estimates and adjusted odds ratios are shown in Appendix A.

A QUEST decision tree was executed. The same variables were considered in the decision tree. Growth was limited to minimum parent node size = 20 and minimum child node size = 5. The tree was pruned to avoid overfitting the model. Only one variable split the tree, which was preeclampsia. Within patients with preeclampsia, 60.8% had preterm delivery. On the other hand, within the node of women who did not develop preeclampsia, only 26% delivered prematurely. Testing negative to COVID-19, positive but asymptomatic, or the C_T_ count did not split the tree. The QUEST tree is shown in Figure 6.

## 4. Discussion

More research is needed in perinatology to fully understand this virus’s behavior during a highly vulnerable period such as pregnancy. Researchers have focused on the SARS-CoV-2 virus’s ability to achieve vertical transmission to the fetus and the participation of the activation of the immune system during COVID-19 as a risk factor for adverse pregnancy and fetal outcomes. By knowing the virus’s pathogenesis during the gestational period and its interaction with the placenta, we may decrease adverse outcomes related to COVID-19 in pregnant women. Pregnancy has, to date, been associated with an increased risk of acquiring respiratory infection with higher morbidity and mortality than the non-pregnant subjects. A proportion of patients with COVID-19 have presented extrapulmonary clinical manifestations, among them cardiac, kidney, liver, digestive tract injuries, and neurological disorders. Placentas are not the exception [20]. Therefore, in this study, we aimed to evaluate placental disorders related to COVID19.

Pregnancy is a period of immunoregulation that allows the fetus to develop in the womb while protecting the mother from infections [21]. The maternal system is complex and governed by multiple hormonal and metabolic factors, including those provided to her via de fetus [22]. SARS-CoV2 modulates not only immune responses and endothelial function but also the coagulation system and tissue repair mechanisms. Thus, the placenta analysis for the presence of fibrinoid, hemorrhage, inflammation, and vasculopathy in pregnant women with COVID-19 is essential in understanding the pathogenesis of the disease during the gestational period. In a recent review, several histomorphology alterations of the placenta have been found in pregnant women infected with SARS-CoV-2 in the second and third trimesters, including inflammation and maternal-neonatal vascular malperfusion [20].

In this study, we further explored the placentas of a group of pregnant women regarding the presence of placental dysmaturity, vasculopathy, fibrinoid, chorangiosis, chorioamnionitis, hemorrhage, or infarction. The placentas of women infected with SARS-CoV-2 had a higher rate of fibrinoid deposition, a clinical feature of maternal vascular malperfusion (MVM), than controls. Moreover, 90% of asymptomatic women with C_T_ values less than 25 developed placental fibrinoid compared to no differences in C_T_ values in symptomatic patients. On this matter, Rebutini et al. found accentuated but non-significant fibrin deposition in placentas of COVID19 positive women compared to non-COVID19 patients [23]. This is somewhat similar to our findings. However, we did not compare COVID19 placentas to healthy pregnancy placentas, but between COVID19 positive women who had low and high C_T_ values, finding fibrinoid significantly higher in placentas from women with lower C_T_ values. Interestingly, all symptomatic women who developed placental fibrinoid had C_T_ counts below 30, and most asymptomatic women with C_T_ counts below 30 developed fibrinoid. The histopathological comparison between placentas from COVID19 positive and negative women was not the objective of this work but between C_T_ values in COVID19 placentas. However, we are currently studying COVID19 and healthy placentas as part of another manuscript in preparation. In contrast, several studies have investigated the placenta of infected mothers and have found the formation of fibrin layers mirroring a massive perivillous fibrin deposition in placentas positive for SARS-CoV-2 RNA by in situ hybridization. Furthermore, SARS-CoV-2 infection of the placenta was shown to be associated with a combination of histopathological features, now termed SARS-CoV-2 placentitis, which include histiocytic intervillositis, perivillous fibrin deposition, and trophoblast necrosis [9,10,24]. Although we did not study SARS-CoV-2 infection of the placenta by in situ hybridization or immunohistochemistry, these findings underpin our results.

Shanes and cols. assessed 16 placentas from COVID19 women and found chronic inflammatory pathology, specifically chronic deciduitis and villitis, atherosis, and fibrinoid necrosis; almost half of the placentas had intervillous thrombi, which has been associated with oligohydramnios, fetal growth restriction, preterm birth, and stillbirth [25]. A review analyzed 29 articles describing the histological changes in the placenta from SARS-CoV-2 positive women [20]. The results show increased MVM in 37.8% of the placentas from SARS-CoV-2 infected women. This is in agreement with our results regarding the presence of changes in placental dysmaturity, vasculopathy, fibrinoid, chorangiosis, chorioamnionitis, hemorrhage, or infarction. In this sense, we have also observed such histopathological changes in the placenta of a first-trimester miscarriage, including active chronic intervillositis accompanied by Hofbauer cell inflammation and hyperplasia [6].

We also aimed to quantify the association between COVID-19 infection during pregnancy and maternal characteristics, adverse outcomes, and Ct values. In the current study, the C_T_ value of less than 37.5 was interpreted as positive for SARS-CoV-2 RNA and further categorized as high C_T_ (>30), medium C_T_ (25–30), or low C_T_ (<25), according to cut-off points used in previous studies [19]. However, this study found no significant association between clinical symptoms and C_T_ values, likely because most patients had mild or no symptoms. In contrast, work from Tanacan and cols. showed increased adverse obstetric outcomes and more extended hospital stays with lower C_T_s (<22.9) [26]. The difference between this work and our results could probably be explained by the population under study, which included pregnant women with severe to moderate symptoms compared to only mild symptoms or asymptomatic patients in our study.

Regarding the association of preeclampsia with SARS-CoV-2 infection, recent multinational studies have shown its association with preeclampsia, especially between nulliparous women and independent of pre-existing factors [27], and have reported an increased risk of RR = 1.76 (95% CI, 1.27–2.43). Moreover, it represents one of the primary indications for preterm delivery [28]. On this matter, we also found an association between COVID19 and preeclampsia but mainly in symptomatic patients and in women who were infected by SARS-CoV-2 before the 35th week of gestation.

Preeclampsia is a disorder related to oxidative stress, endothelial damage, and the presence of anti-antiangiogenic factors [29]. Preeclampsia also has overlapping symptomatology with COVID19, such as endothelial damage, multi-organ failure, coagulopathy, and liver injury, which may be why the possibility of an association between these two disorders was likely to occur. However, Mendoza et al. propose that pregnant women with severe COVID19 may develop a preeclampsia-like syndrome and not the actual preeclampsia that is related to placental alterations because they found normal sFlt-1/PlGF, UtAPI, and LDH levels in all but one of the five women with severe COVID19 who presented signs and symptoms of preeclampsia [30]. No angiogenic nor anti-angiogenic factors were measured in this study, but we did find several placental alterations in women who developed COVID-19 during pregnancy.

### Strengths and Limitations of the Study

One of the strengths of this study is the examination of the placentas of the women enrolled. Few studies have included the pathological assessment of a considerable number of placentas from pregnant women who developed COVID-19.

Nonetheless, this study had several limitations. First, it is a retrospective study in which patients from only one perinatology hospital in Mexico were studied. Thus, its generalizability may be questioned. Second, an institutional database was used, and many offspring variables were not analyzed for this specific study. Third, the patients attending the National Institute of Perinatology are mainly patients with high-risk pregnancies and comorbidities, e.g., diabetes, obesity, autoimmunity, transplants, and cardiopathy. Again, this may affect the generalizability of the study results. Fourth, C_T_ values may vary depending on the method of specimen collection and platform. Additionally, no standard exists to validate quantitative assays between laboratories worldwide. Fifth, the placental pathology should have been compared to normal placentas. In this study, we only had placentas from COVID19 positive women. Thus, we only compared the presence of placental alterations between different levels of C_T_ counts and symptomatology. The comparison with normal placentas would have given us a clearer perspective.

## 5. Conclusions

COVID-19 in pregnant women was associated with preeclampsia but not preterm birth, and in both cases, no differences in C_T_ counts were found. Regarding placental pathology, the presence of fibrinoid was uncovered in women infected with SARS-CoV-2 and low PCR C_T_ values. This evidence supports the relative importance of C_T_s and placental health during a SARS-CoV-2 infection and requires further investigation.

## Figures and Tables

**Figure 1 viruses-13-01884-f001:**
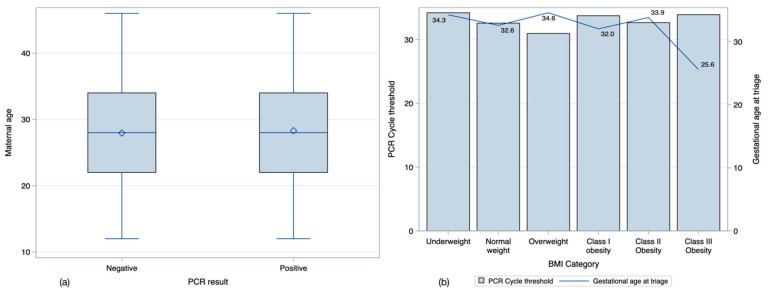
Age and BMI characteristics. Maternal age was similar between women who tested positive and negative to COVID (**a**). Women who were overweight tended to have lower C_T_ values, and women with class three obesity had the lowest gestational age at triage (**b**).

**Figure 2 viruses-13-01884-f002:**
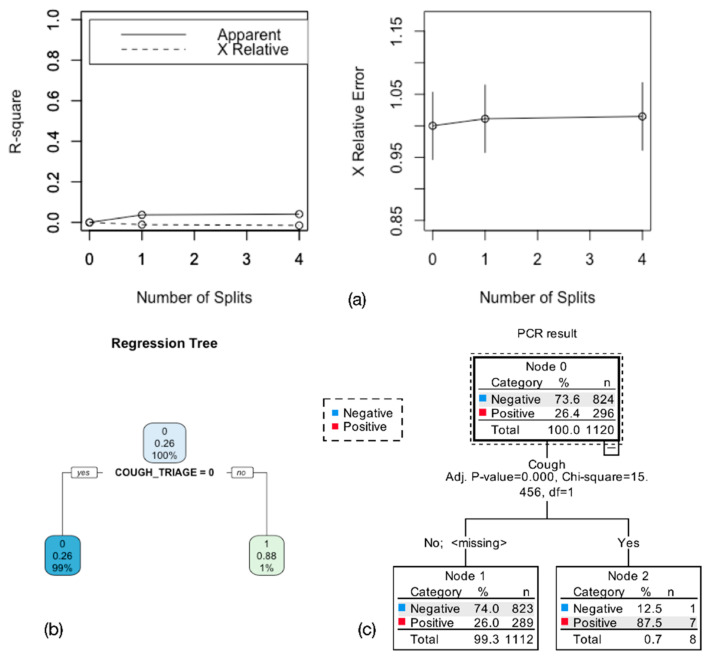
Regression tree for symptom variables (cough, fever, headache, dyspnea, myalgias, rhinorrhea, diarrhea, thrombosis, neurological symptoms, arrhythmia, lethargy) and PCR test positivity. (**a**) shows the cross-validated error of the regression tree. Only one variable split the tree, indicating that 88% of women who tested positive had cough at triage (**b**). CHAID tree also had one split, and the terminal nodes indicate that cough was present in 88% of women who tested positive, whereas only 26% of those who tested positive did not have cough (**c**).

**Figure 3 viruses-13-01884-f003:**
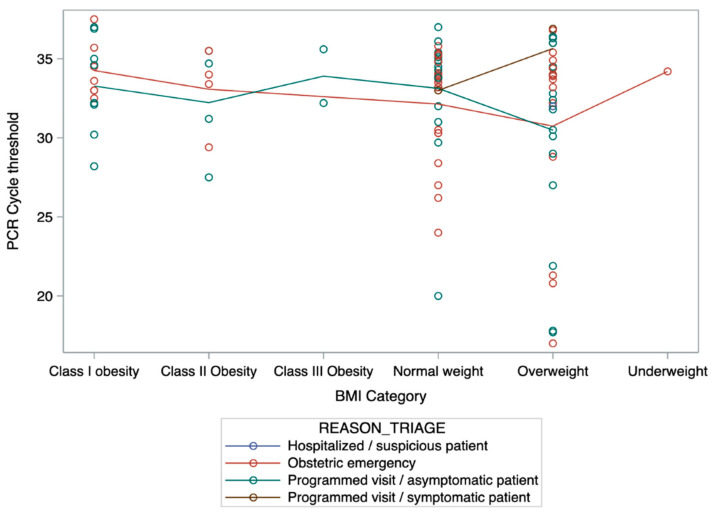
Two-way ANOVA C_T_ distribution between BMI categories. Women with overweight tended to have lower (non-significant) C_T_ values.

**Figure 4 viruses-13-01884-f004:**
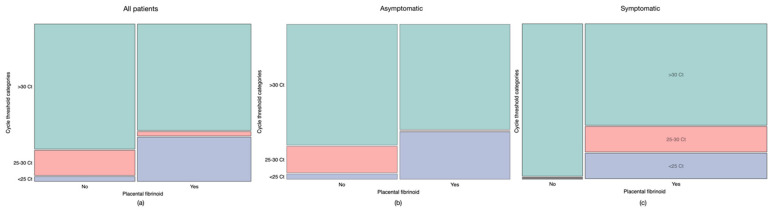
Presence of placental fibrinoid according to C_T_ values. (**a**) all patients with a positive PCR test, (**b**) asymptomatic patients, (**c**) symptomatic patients. Lower C_T_ values were more common in placentas from women who developed fibrinoid (**a**). asymptomatic patients who developed fibrinoid had either low or high C_T_s (**b**). All women with low C_T_ values (<25), who were symptomatic, developed fibrinoid (**c**).

**Figure 5 viruses-13-01884-f005:**
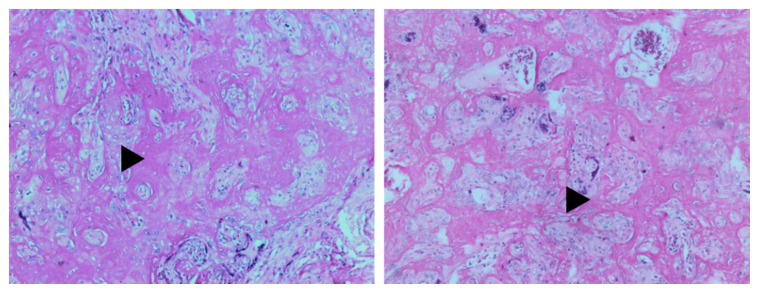
Presence of placental fibrinoid. The hematoxylin and eosin-stained photomicrograph of the placenta shows the chorionic villi encased by massive perivillous fibrin deposition (arrowheads). Original magnification ×20.

**Figure 6 viruses-13-01884-f006:**
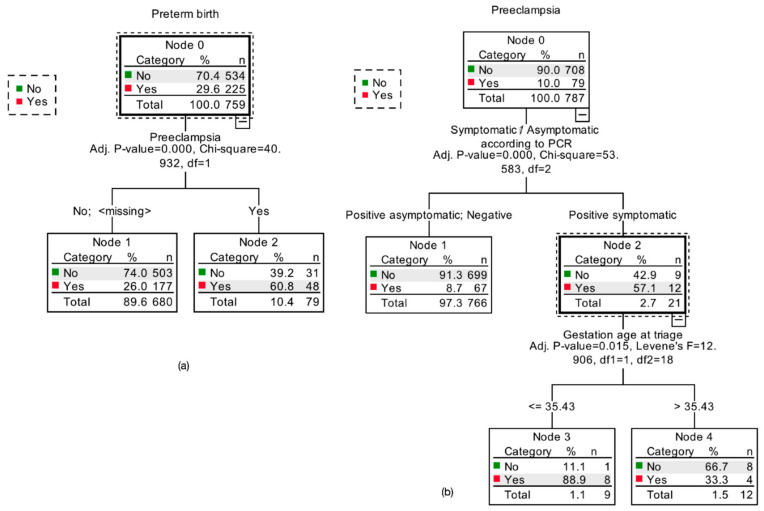
QUEST trees for preterm birth (**a**) and preeclampsia (**b**). QUEST for preterm birth had only one split. Preterm birth was highly related to the development of preeclampsia; the terminal node of the tree shows that 60.8% of women who developed preeclampsia had preterm birth, whereas in women who did not develop preeclampsia, only 26% delivered preterm (**a**). QUEST for preeclampsia had two splits. Preeclampsia was present in 8.7% of women who tested negative or were positive asymptomatic, but 57.1% of positive and symptomatic women developed preeclampsia. Moreover, the terminal node shows that women who developed COVID before or at week 35 and who were symptomatic were more likely to develop preeclampsia (88.9%).

**Table 1 viruses-13-01884-t001:** Maternal characteristics.

Characteristics		COVID-19 Positive (298) Mean ± SD N (%)	COVID-19 Negative (828) Mean ± SD N (%)	All Patients (1126) Mean ± SD N (%)	*p*
Maternal age		27.94 ± 7.2	28.3 ± 7.4	28.04 ± 7.3	0.48
Pre-pregnancy BMI		27.05 ± 5.4	27.17 ± 5.8	27.12 ± 5.2	0.74
Weight gain per week		0.09 ± 0.52	0.20 ± 0.22	0.17 ± 0.35	0.08
Pre-pregnancy BMI category:	underweight	1 (12.5)	7 (87.5)	8	>0.05
	normal weight	38 (36.9)	65 (87.5)	103	>0.05
	overweight	38 (33)	77 (67)	115	>0.05
	class I obesity	15 (36.6)	26 (63.4)	41	>0.05
	class II obesity	8 (40)	12 (60)	20	>0.05
	class III obesity	2 (20)	8 (80)	10	>0.05
Number of pregnancies		2.19 ± 1.3	2.23 ± 1.3	2.22 ± 1.33	0.14
Number of deliveries		0.41 ± 0.8	0.44 ± 0.8	0.43 ± 0.8	0.46
Number of c-sections		0.57 ± 0.82	0.48 ± 0.75	0.51 ± 0.77	0.03
Number of miscarriages		0.42 ± 0.77	0.50 ± 0.90	0.48 ± 0.87	0.01
Diabetes	No	257 (25.8)	740 (74.2)	997	>0.05
	Type 1 Diabetes	3 (30)	7 (70)	10	>0.05
	Type 2 Diabetes	11 (24.4)	34 (75.6)	45	>0.05
	Gestational diabetes	15 (33.3)	30 (66.7)	45	>0.05
Chronic hypertension	No	260 (26.3)	729 (73.8)	989	>0.05
	Yes	12 (23.5)	39 (76.5)	51	>0.05
Autoimmunity (SLE, RA)	No	262 (25.5)	756 (74.5)	1027	>0.05
	Yes	15 (33.3)	30 (66.7)	45	>0.05
Asthma	No	287 (26.1)	811 (73.9)	1098	>0.05
	Yes	11 (68.8)	5 (31.3)	16	>0.05
Cardiopathy	No	288 (26.3)	807 (73.7)	1095	>0.05
	Yes	4 (21.1)	15 (78.9)	19	>0.05
Hypothyroidism	No	157 (25.4)	460 (74.6)	617	0.001
	Yes	18 (51.4)	17 (48.6)	35	0.001

BMI, Body Mass Index; SLE; Systemic Lupus; RA, Rheumatoid arthritis. T student was used for differences in numerical variables and chi-square, or Fisher tests were performed to assess differences in proportions.

**Table 2 viruses-13-01884-t002:** Clinical characteristics at triage.

Characteristics		COVID-19 Positive (298) Mean ± SD N (%)	COVID-19 Negative (828) Mean ± SD N (%)	All Patients (1126) Mean ± SD N (%)	*p*
Gestational age at triage		33.40 ± 8.3	33.35 ± 8.3	33.37 ± 8.4	0.91
Temperature at triage		36.31 ± 0.49	36.35 ± 0.48	36.34 ± 0.48	0.58
Breathing rate at triage		19.01 ± 3.3	19.4 ± 5.4	19.12 ± 3.9	0.07
Heart rate at triage		79.01 ± 13.2	78.28 ± 10.1	78.46 ± 11	0.008
O_2_ saturation at triage		94.82 ± 1.9	94.43 ± 1.7	94.61 ± 1.8	0.73
SBP at triage		112.2 ± 16.4	110.47 ± 13	110.9 ± 14	0.07
DBP at triage		70.41 ± 9.6	69.98 ± 8.7	70 ± 8.9	0.03
Triage distribution of patients	Programmed visit/asymptomatic patient	131(28.7)	325 (71.3%)	456	>0.05
	Programmed visit/symptomatic patient	8 (40)	12 (60%)	20	>0.05
	Obstetric emergency	140 (24.9)	422 (75.1)	562	>0.05
	Hospitalized/suspicious patient	15 (19.2)	63 (80.8)	78	>0.05
Women with a positive test and at least one symptom during evaluation	Asymptomatic Symptomatic	90.2 (269) 9.8 (29)		298	<0.001
Decision after test	Ambulatory treatment	81 (37.9%)	133 (62.1%)	214	<0.001
	Hospital admission	216 (23.6)	696 (76.4%)	912	<0.001

**Table 3 viruses-13-01884-t003:** Pregnancy outcomes.

Characteristics		COVID-19 Positive (298) Mean ± SD N (%)	COVID-19 Negative (828) Mean ± SD N (%)	All Patients (1126) Mean ± SD N (%)	*p*
Week at delivery		37.5 ± 3	36.44 ± 3.7	37 ± 3.4	0.28
Hospitalization days at delivery		1.72 ± 1.8	1.76 ± 1.8	1.75 ± 1.3	0.13
Pregnancy resolution	Uterine curettage	9 (22.5)	31 (77.5)	40	>0.05
	Forceps delivery	3 (37.5)	5 (62.5)	8	>0.05
	Eutocic delivery	64 (26.7)	176 (73.3)	240	>0.05
	C-section	124 (25.2)	368 (74.8)	492	>0.05
	Ectopic pregnancy	0	1 (100)	1	>0.05
	Manual vacuum aspiration	0	6 (100)	6	>0.05
Pregnancy hypertensive disorders	No	173 (24.5)	533 (75.5)	706	>0.05
	Gestational hypertension	3 (37.5)	5 (62.5)	8	>0.05
	Preeclampsia	8 (30.8)	18 (69.2)	26	>0.05
	Preeclampsia with severity features	16 (34.8)	30 (65.2)	46	>0.05
	HELLP	1 (20)	4 (80)	5	>0.05
	Eclampsia	0	1 (100)	1	>0.05
Premature rupture of membranes	No	177 (25.8)	510 (74.2)	687	>0.05
	Yes	23 (22.3)	80 (77.7)	103	>0.05
Preterm birth	No	161 (25.9)	461 (74.1)	622	>0.05
	Yes	39 (23.2)	129 (76.8)	168	>0.05
Obstetric hemorrhage	No	193 (25.7)	559 (74.3)	752	>0.05
	Yes	7 (18.4)	31 (81.6)	38	>0.05
Fetal death	No	190 (25.1)	566 (74.9)	756	>0.05
	Yes	19 (33.3)	38 (66.7)	57	>0.05
Newborn weight (grams)		2672.29 ± 823	2661.43 ± 836	2666.9 ± 825	0.83
Newborn head circumference (cm)		33.24 ± 2.7	32.63 ± 2.9	32.9 ± 2.9	0.93
Apgar 1		7 ± 2.2	6.5 ± 2.4	6.79 ± 2.3	0.08
Apgar 5		8.36 ± 1.7	8.33 ± 1.5	8.35 ± 1.7	0.85

## Data Availability

Data supporting reported results can be obtained from the authors upon request due to ethics.

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
