# Peer review of "COVID-19 Infection in Pregnancy: PCR Cycle Thresholds, Placental Pathology, and Perinatal Outcomes"

_viruses, 2021, doi:10.3390/v13091884_

Round 1

Reviewer 1 Report

The manuscript is timely and well written. I urge the authors to address these suggestions/corrections and resubmit:

  1. New data has come out in regards to the specific features of SARS-CoV-2 placentitis (see doi.org/10.5858/arpa.2021-0246-SA and doi.org/10.5858/arpa.2021-0164-ed.  The authors should address these specific findings in their study if possible.
  2. It would be most interesting to know if any of the placentas were infected. Could the authors study at least the ones with "fibrinoid" by RNA-ISH or immunohistochemistry for this?
  3. Were any of the births, live or still, infected?
  4. A figure of the pathology of "fibrinoid" should be included.
  5. Was the weight percentile of the placentas analyzed? Were these placentas in general small? Did that correlate with CT values?
  6. Line 313 severity should read severe
  7. Lines 315-317 these were all the livebirths, did the stillbirth weights differ by symptoms or CT values? What were the causes of the stillbirths? Placental?
  8. The results described in lines 313-315 are different from what is presented in Table 3.
  9. What was the time interval between the positive RT-PCR test and delivery? Were these all active infections or are some of them recovered?

The discussion is a little long but reads well. The strengths and limitations section is well written. It is hard to get around  some of the weaknesses but I don't think they are fatal. 

Author Response

We are thankful for the reviewer’s comments that have considerably improved the manuscript. Below you may find the changes that were made to the article, following your comments and suggestions.

Reviewer 1.

The manuscript is timely and well written. I urge the authors to address these suggestions/corrections and resubmit:

  1. New data has come out in regards to the specific features of SARS-CoV-2 placentitis (see doi.org/10.5858/arpa.2021-0246-SA and doi.org/10.5858/arpa.2021-0164-ed.  The authors should address these specific findings in their study if possible.

We have addressed these findings in the Introduction and Discussion section (lines 80-82 and lines  511-518, respectively).

  1. It would be most interesting to know if any of the placentas were infected. Could the authors study at least the ones with "fibrinoid" by RNA-ISH or immunohistochemistry for this?

This is a very interesting comment however, we do not have tissue of these placentas to perform either RNA-ISH or immunohistochemistry. However, we collected from the electronic medical record the result of the SARS-CoV-2 RT-qPCR of these placentas. Thirty-two out of the 66 placentas had a PCR test result; of those 15 placentas were positive. We first executed a ROC curve with maternal Ct values to obtain a cut-off point for placental positivity and found a significant association between maternal Ct values below 33 with a positive placental test (p=0.021). This is now indicated in lines 351-357 of the manuscript.

To further characterize if the presence of fibrinoid could be associated with placental Ct values, we performed a ROC curve and found a trend for a higher number of fibrinoid in positive placentas but not a significant association (p=0.049). Although we acknowledge the number of placentas to be small. This is now indicated in lines 393-400 of the manuscript.

  1. Were any of the births, live or still, infected?

Twenty-seven live births were confirmed to be infected by PCR test at newborn follow-up. This data is part of another study and will be published in the near future.

  1. A figure of the pathology of "fibrinoid" should be included.

We have included in lines 375-383 a representative figure of “fibrinoid” showing a massive perivillous fibrin deposition (Figure 5).

  1. Was the weight percentile of the placentas analyzed? Were these placentas in general small? Did that correlate with CT values?

Placental percentiles were calculated for all 66 placentas according to Almog et al. (DOI: 10.1016/j.placenta.2010.10.008. Placentas were, in the majority, small. 50.8% were equal or less than percentile 3. This is now indicated in lines 344-346. A crosstabs analysis showed no significant association between percentile categories and Ct categories. No significant correlation was found between the Ct numerical variable and the placental weight percentile variable.

  1. Line 313 severity should read severe.

The word “severity” was used to refer to preeclampsia with severity features throughout the manuscript.

  1. Lines 315-317 these were all the livebirths, did the stillbirth weights differ by symptoms or CT values? What were the causes of the stillbirths? Placental?

We have described if placentas were from livebirths, abortion or miscarriages, and performed chi-square analysis to assess if placental size differed according to Ct value categories. One of the two miscarriages was related to multiple fetal malformations. This information is now included in lines 341-344.

  1. The results described in lines 313-315 are different from what is presented in Table 3.

Table 3 shows pregnancy outcomes of 298 COVID19 positive and 828 negative women.  However, only 66 placentas out of the 298 positive pregnant women from this same cohort were evaluated by a pathologist.

  1. What was the time interval between the positive RT-PCR test and delivery? Were these all active infections or are some of them recovered? The discussion is a little long but reads well. The strengths and limitations section is well written. It is hard to get around  some of the weaknesses but I don't think they are fatal. 

The time interval between the RT-PCR test and delivery was 24-48 h depending if the pregnant woman was admitted for an appointment for delivery, a programmed visit 24-48h before delivery or if it was an obstetric emergency, in which case the test is performed upon arrival (test result at 24h). Nevertheless, the majority of patients were tested upon their admittance for an obstetric emergency (562 women). Therefore, they were all active infections. We described this in the Methods section (lines 126-130).

Reviewer 2 Report

The manuscript by Laresgoiti-Servitje et al aims to compare maternal, fetal, and placental outcomes with CT results from the maternal SARS-CoV-2 nasal swabs. This is a very significant topic, but important points must be addressed.

First, as part of the medical/scientific community, we need to stop equating Ct values to viral load. A Ct value taken from an RT-PCR machine can tell you the relative abundance of a gene (is it there or is it not there) compared to a known standard gene (housekeeping gene or preset nucleic acid standard concentration). CT values are not comparable between tests or between different lots of the same test because they are dependent on numerous factors such as the specimen collection, storage, transport, time from collection, nucleic acid target, primers and probes, extraction method, amplification method, instruments used, etc. There is no association between Ct values and viral load, viral transmissibility, or symptoms and many papers have already demonstrated this (Tian E et al 2021; Mathers AJ 2020 off the top of my head). The only things we are doing by continuing to publish that Ct=viral load is confusing the public and damaging scientific integrity. Therefore, the authors must remove any mention of viral load from the manuscript because that is not what they have measured and report their results in the context of low and high Ct values, which is perfectly acceptable and sound. 

Can the authors include a methods section on the RT-PCR assay?  

Line 177 – could the authors clarify the value BMI they considered most patients to have? BMI >30kg/m2?

Figure legends should be more descriptive. I have a difficult time understanding what the authors are trying to show in the figures within the context of their results.

For the placenta section, only 66 placentae out of the 298 covid positive patients in table 1 are mentioned. Are these the same patients or a different cohort? When were they infected in the context of delivery/pathological exam and why did they have an exam when the others did not? Please describe this cohort more clearly in the text.

Line 290 -Could the authors indicate the number of subjects in each of the Ct value groups? Percentages can be very deceptive when working with small numbers.

Table 1 lists 298 subjects where table 3 lists 299. Which is correct?  

Figure 5 is mislabeled in the figure legend and it is not clear what a and b indicates. Additionally, the decision tree figure is very raw, is there a way to make this figure more polished? It is difficult to understand the significance of this image as is.

Author Response

Reviewer 2.

The manuscript by Laresgoiti-Servitje et al aims to compare maternal, fetal, and placental outcomes with CT results from the maternal SARS-CoV-2 nasal swabs. This is a very significant topic, but important points must be addressed.

First, as part of the medical/scientific community, we need to stop equating Ct values to viral load. A Ct value taken from an RT-PCR machine can tell you the relative abundance of a gene (is it there or is it not there) compared to a known standard gene (housekeeping gene or preset nucleic acid standard concentration). CT values are not comparable between tests or between different lots of the same test because they are dependent on numerous factors such as the specimen collection, storage, transport, time from collection, nucleic acid target, primers and probes, extraction method, amplification method, instruments used, etc. There is no association between Ct values and viral load, viral transmissibility, or symptoms and many papers have already demonstrated this (Tian E et al 2021; Mathers AJ 2020 off the top of my head). The only things we are doing by continuing to publish that Ct=viral load is confusing the public and damaging scientific integrity.

  1. The authors must remove any mention of viral load from the manuscript because that is not what they have measured and report their results in the context of low and high Ct values, which is perfectly acceptable and sound. 

We agree with the reviewer and appreciate the kind comments. The term viral load was removed throughout the manuscript and only the Ct values are reported instead.

  1. Can the authors include a methods section on the RT-PCR assay?  

We have included the RT-qPCR assay (lines 145-1).

  1. Line 177 – could the authors clarify the value BMI they considered most patients to have? BMI >30kg/m2?

Patients who were considered as overweight had a BMI between 25-29.9 and those considered as obese, BMI ≥30. It has been clarified in the manuscript (lines 211-212).

  1. Figure legends should be more descriptive. I have a difficult time understanding what the authors are trying to show in the figures within the context of their results.

Thank you for the suggestion. We have explained more thoroughly what the figures show.

  1. For the placenta section, only 66 placentae out of the 298 covid positive patients in table 1 are mentioned. Are these the same patients or a different cohort? When were they infected in the context of delivery/pathological exam and why did they have an exam when the others did not? Please describe this cohort more clearly in the text.

Yes, the 66 placentas are the same patients 298 positive patients mentioned in Table 1. The characteristics of the mothers are now described in the manuscript.

The time interval between the RT-PCR test and delivery was 24-48 h depending if the pregnant woman was admitted for an appointment for delivery, i.e. a programmed visit 24-48h before delivery or if it was an obstetric emergency, in which case the test was performed upon admittance and the test result was obtained between the next 24h. Nevertheless, the majority of patients were tested upon their admittance for an obstetric emergency (562 women), the rest were evaluated during their programmed control visits 24-48h prior to delivery (this is indicated in lines 126-130). The pathological exam of the placenta was performed between the first 24h after birth, at the same time interval as the RT-PCR test.

Submitting all placentas for pathological examination is not clinically indicated. The placental examination was recommended for women with a positive RT-qPCR test however, not all of the placentas were routinely obtained and sent to the Department of Anatomic Pathology. This was included in the Methods sections of the manuscript (lines 130-141).

  1. Line 290 -Could the authors indicate the number of subjects in each of the Ct value groups? Percentages can be very deceptive when working with small numbers.

The number of pregnant women that correspond to the percentage was added in lines 315-320 as suggested by the reviewer.

  1. Table 1 lists 298 subjects where table 3 lists 299. Which is correct?  

We appreciate the observation, 298 is the correct number. It has been changed in the manuscript.

  1. Figure 5 is mislabeled in the figure legend and it is not clear what a and b indicates. Additionally, the decision tree figure is very raw, is there a way to make this figure more polished? It is difficult to understand the significance of this image as is.

Figure legend was changed to broadly explain what the trees are showing and what those percentages mean. This figure is now depicted as Figure 6 since we have included a new figure of fribrinoid (Figure 5).

Round 2

Reviewer 1 Report

Thanks for the response to  my comments and suggestions.

Reviewer 2 Report

The authors did a fantastic job addressing concerns and clarifying the placenta population more clearly. Additionally, the new and improved figures are very well done.